# Temporomandibular Joint Facts and Foibles

**DOI:** 10.3390/jcm12093246

**Published:** 2023-05-01

**Authors:** Louis Gerard Mercuri

**Affiliations:** Department of Orthopaedic Surgery, Rush University Medical Center, Chicago, IL 60612, USA; louis_g_mercuri@rush.edu

**Keywords:** temporomandibular joint, TMJ, temporomandibular joint disorders, TMD

## Abstract

The purpose of this article is to dispel some of the major foibles associated with the etiology and management of TMJ disorders, while presenting some of the facts based on the scientific literature to date. To appreciate this kind of update, the reader must be an “out of the box thinker” which requires openness to new ways of seeing the world and a willingness to accept new concepts based on evolving evidence.

## 1. Introduction

In a 24 August 2015 New York Times op-ed entitled “The Case for Teaching Ignorance”, Jamie Holmes wrote, “Presenting knowledge as more certain than it is discourages curiosity”. There is no healthcare issue in dentistry that has proffered more quasi-scientific information than what is related to the etiology and management of temporomandibular joint (TMJ) disorders.

In a 1995 literature review commissioned by NIDR (now NIDCR) to determine the strength of the evidence regarding therapy for TMJ disorders, Antczak-Bouckoms found that the TMJ literature as of that date consisted primarily of uncontrolled clinical trials, case series, case reports, and simple descriptions of techniques; she concluded that such uncontrolled observations were subject to considerable bias and thus difficult to interpret. Further, she reckoned that “if treatment of TMD is going to follow the evidence-based trend in medicine, rather than opinion or pathophysiologic rationales, then more rigorously controlled clinical trials of most therapeutic options would be necessary” [1]. However, while much has changed, there are still a number of “uncontrolled clinical trials, case series, case reports, and simple descriptions of techniques” in the present day TMJ literature.

The purpose of this article is to dispel some of the major foibles associated with the etiology and management of TMJ disorders, while presenting some of the facts based on the scientific literature to date. To appreciate this kind of update, the reader must be an “out of the box thinker” which requires openness to new ways of seeing the world and a willingness to accept new concepts based on evolving evidence.

## 2. The Temporomandibular Joint

What is the TMJ besides being the articulation between the mandible and the base of the skull that allows the functions of mastication and speech as well as supporting deglutition and the upper airway? Do we want people to utilize cynical descriptions of the TMJ as being The Mouth Joint, The Money Joint, or The Mystery Joint? For surgeons, why is treating TMJ problems like looking for a good restaurant in an unfamiliar town where you do not know which joint to enter and which one to stay out of? (Seldin, E 1983).

As I see it, there are 10 foibles related to the TMJ that I feel must be addressed and investigated further, but even now there are many evidence-based facts to counteract them.

### 2.1. Problem #1

Patients, physicians, many dentists, and health care insurance providers do not consider the TMJ as being just another orthopedic joint. Orthopedics is the branch of medicine concerned with the diagnosis and treatment of acute, chronic, traumatic, and overuse injuries and other disorders of the musculoskeletal system. Is that not exactly what clinicians do who manage TMJ patients?

### 2.2. Problem #2

The etiology, diagnosis, and management focus has often been on only one joint component (e.g., the disc, masticatory muscles, teeth). The musculoskeletal system is a complex scheme made up not just of the bone and cartilage of the joints, articular discs, and menisci, but also the muscles that move the joints as well as the vascular, lymphatic, and neural components that nourish the joint components and prompt its movements [2].

### 2.3. Problem #3

Unfortunately, the TMJ has been compared anatomically to the hip, a ball-in-socket joint, as opposed to the knee, which has a similar range of motion with both sliding and lateral movements. Both joints have an interposed fibrocartilaginous structure, anatomically designated a “meniscus” in the knee, but described as a “disc” in the TMJ due to its anatomic shape and the nature of the supporting attachments. Intra-articular fibrocartilages are complete or incomplete plates of fibrocartilage that are attached to the joint capsule (the investing ligament) and that stretch across the joint cavity between a pair of conarticular surfaces. When complete, they are called “discs”; when incomplete, they are called “menisci” [3].

### 2.4. Problem #4

The etiology, diagnosis, and management focus for TMJ disorders has recently been fixated on the position of the disc. At first it was all about the occlusion of the teeth or the malposition of the mandibular condyle. However, when those mechanistic concepts were disproven, “internal derangement” of the articular disc became the target for both non-surgical and surgical treatments.

The term “internal derangement” was coined in 1814 by an English surgeon William Hey for meniscus problems in the knee [4]. The term was applied to the TMJ disc in 1826 by another English surgeon, Astley Cooper [5]. In 1887, Thomas Annandale, a Scottish surgeon, reported the repositioning of an “internally deranged” TMJ disc with horsehair sutures [6]. Soon thereafter, TMJ disc internal derangement became the prime suspect for joint noise, pain, and “locking”, and this concept became the basis for performing TMJ discectomy [7].

Etiologic concepts of TMJ disorders shifted from these early versions of internal disc derangement to a strong emphasis on the teeth and muscles. This continued until the 1970s, when William Farrar and William McCarty developed the Normandy Study Group and resurrected internal derangement of the TMJ disc as being the main anatomical etiology of TMJ disorders [8].

Clyde Wilkes went a step further by stating that disc displacement led directly to condylar degeneration. Wilkes then proposed a classification and surgical management procedures for repositioning the disc [9]. This hypothesis led to the belief that internal derangements, or disc displacements, represented the basic pathological entity responsible for all the clinical signs and symptoms associated with the so-called TMJ pain dysfunction syndrome.

Clinical management of TMJ disorders took on new importance for surgeons because it was now assumed, based on Wilkes’ assertions, that disc displacement was likely to progress to advanced degenerative states. This prompted the development of open surgical disc repositioning protocols by Dolwick [10], arthroscopic protocols by Sanders [11], McCain [12], and Yang [13], as well as arthrocentesis protocols by Nitzan [14], all of which added to further understanding of the role of the TMJ disc in healthy and diseased states.

The author Dan Brown said, “Wide acceptance of an idea is not proof of its validity” [15]. This is true of many things, especially TMJ “internal derangement” and its progression to degeneration of the articulation.

De Bont stated that “disc displacement may represent extremes of normal variation” and “…internal derangement may be a sign of a range of conditions, rather than a single entity” [16].

Widmalm reported that “internal derangement does not always cause clinical symptoms; it may be contributory, but inflammatory changes in the joint capsule, synovial tissues and retro-discal tissues are likely more important for the development of symptoms than disc displacement or deformation per se [17]”.

With the advent of MR imaging, investigators began to study the TMJs of random asymptomatic volunteers and found that 23–35% of these subjects had displaced discs [18,19,20,21,22,23,24,25,26]. This was analogous to studies of intervertebral disc displacement in asymptomatic low back pain subjects [27,28].

Since 23–35% of the population have abnormal disc–condyle relationships, it can be argued that MRI disc position does not have a direct correlation with clinical signs and symptoms. Thus, there is a need to understand how much of the biological variability of the disc–condyle relationship is really part of a natural course of joint wear and aging, and how much of it is the result of joint overload. A multivariate analysis did not show that TMJ pain correlates with a reduction in anterior disc displacement or mandibular condyle morphology [29].

Joint biomechanical instability is defined as the inability to maintain the functional relationship between the bones and associated soft tissues under normal physiological forces. When this occurs, it puts extra stress on the intra-articular and surrounding joint structures, leading to soft tissue damage or tears, joint degeneration, and pain [30]. Joints begin to degenerate or break down when the catabolic (destructive) processes exceed the anabolic (reparative) processes. The result is joint instability and disabling chronic musculoskeletal pain.

Histological examinations have confirmed that increased vascularity and inflammatory reactions are seen in the synovia and posterior disc attachment in patients who present with TMJ pathology. Synovial fluid is produced by fibroblast-like type B synovial cells. Physiologic changes in synovial fluid volume and content occur in response to trauma, inflammation, and bacterial, fungal, or viral penetrance. These changes in synovial fluid can increase friction, which can lead to joint degeneration [31,32].

In 2016, Israel concluded that “internal derangement” of the TMJ is not a disease, but a non-specific sign of tissue failure which, in some cases, may lead to biomechanical instability of the joint. This tissue failure is usually caused by TMJ mechanical overloading, leading to inflammatory/degenerative changes related to synovitis. The intra-articular changes associated with internal derangement can also be caused by a systemic arthropathy or a localized atypical arthropathy involving the TMJ [33].

Impairment of joint movement after repeated micro- or macro-trauma, including the TMJ, is the result of the development of synovitis within the joint capsule and Hilton’s Orthopedic Law, i.e., “The nerves that innervate a joint, also innervate the muscles that move that joint, as well as the overlying skin” [34]. In the case of the TMJ, it is the trigeminal nerve (CN V). This reflex limits joint movement, thereby causing pain and possibly further damage.

Mobilization of the disc is of more importance for the reduction in signs and symptoms of internal derangement than anatomically repositioning it in relationship to the condylar head and the articular eminence. Therefore, disc function is more important than disc position. Scientific evidence regarding the effectiveness of TMJ disc repositioning remains scarce and needs further efforts to guide clinicians and patients when considering the clinical and surgical options to treat TMJ disc displacement [35].

Chantaracherd et al., in a cross-sectional study, found no association between TMJ disc status and TMD symptomatology as represented by pain, jaw function, and disability [36]. This suggests that TMJ disc displacements per se have minimal impact on patients’ reported pain, function, and disability. This also suggests that treatments focusing on “correcting” those displacements (such as surgery) may have limited impact on patient-reported outcomes [37].

### 2.5. Problem #5

Dentists are taught to “cure” rather than “manage”. If there is caries and it is removed and replaced with a restoration, the tooth is cured. If there is pulpal disease and endodontics is successfully performed, the tooth is cured. If there are non-restorable caries or advanced periodontal disease and the tooth is extracted, then those problems are cured.

However, there are few if any medical problems that are “cured”; instead, most are “managed”. Hypertension, diabetes, musculoskeletal disorders, etc., are managed, not cured. TMJ disorders are musculoskeletal conditions, and therefore not “curable”, but most of them are “manageable”. This is an important concept for the clinician to accept and pass on to their TMJ patients so as not to give the patient the undeliverable and confounding expectations of a “cure” instead of the realistic and deliverable expectation of a “management” plan for their TMJ problem.

### 2.6. Problem #6

Patients, physicians, many dentists, and health care insurers are often using the wrong terminology. The commonly used term “Temporomandibular Disorders” (TMDs) is a collective label that is used to embrace several clinical problems that involve the masticatory musculature and the temporomandibular joint itself [38]. Therefore, like all musculoskeletal diseases, there are two major categories, namely, extra-articular muscle-based disorders and intra-articular joint-based disorders.

Most patients with signs and symptoms of an extra-articular muscle based TMJ disorder have masticatory and cervical myositis or myofascial pain that is amenable to non-invasive management. However, those with intra-articular TMJ disorders display demonstrable signs and symptoms of specific TMJ pathology, like other joints in the body, some of which can be managed conservatively while others may have indications for invasive management.

### 2.7. Problem #7

There are multiple confusing and often unrelated signs and symptoms that are reported by TMD patients, in addition to the frequent presence of other co-morbid painful conditions.

In 1985, Rugh and Solberg stated “…there is scant evidence to suggest that TMJ conditions are new diseases…rather, it appears that the recent interest is a result of redefining of old conditions, increasing education of professionals, increased public awareness, and a belief by clinicians that these signs and symptoms should be treated.” [39]

As discussed in Problem #4, this kind of thinking led to the belief that disc displacement treatment is necessary to prevent progression of internal derangement disorders to degenerative joint disease. However, several studies have shown that TMJ disorders can often turn out to be self-limiting, or non-progressive conditions [40].

### 2.8. Problem #8

The dental profession has embraced the concept that the TMJ is a unique articulation because it has terminal ectodermal structures (teeth) and has fibrocartilage rather than hyaline cartilage covering the articular surfaces of the condyles and eminence. Therefore, dentistry focused diagnostic and therapeutic modalities on the occlusion and more recently focused on the intra-articular disc position.

The axial skeleton also has terminal ectodermal structures—finger and toenails. Is there any literature that directly correlates in the appendicular degenerative joint changes with abnormalities in finger or toenails?

Manfredini has written that multiple research findings support the absence of a disease-specific association between TMJ disorders and dental occlusion. There seems to be no solid ground to further hypothesize a role for dental occlusion in the pathophysiology of TMJ disorders. He encouraged clinicians to abandon the old gnathological paradigm in the management of TMJ disorders in practice [41].

The essential life functions of mastication, speech, airway support, and deglutition are supported by TMJ function and form. This places the TMJ complex under more cyclical loading and unloading than any other body joint over a lifetime. Therefore, the TMJ is considered to be load-bearing during masticatory function and even at rest or during full closure of the mouth. The fibrocartilaginous tissues, including the disc and articular cartilage, have important functions in stress distribution. Fibrocartilage is the strongest kind of cartilage, which is why the TMJ has fibro- rather than hyaline cartilage (the weakest of the three types of cartilage). Further, the bony surfaces of other highly stressed joints such as the sternoclavicular and symphysis pubis are covered with fibrocartilage [42,43,44].

### 2.9. Problem #9

Sophisms, bad science, opinions, and cults have developed around the diagnosis and management of TMJ disorders.

Why do some adhere to concepts that have proven to be scientifically invalid? When it comes to TMJ disorders, Mohl and Ohrbach believed the following were in play: early professional training and experience, reinforcement of familiar procedures, inertia, isolation in a private practice setting, insecurity, unfamiliarity with the literature, inability to assess scientific evidence, blind belief in “schools of thought” sponsored by charismatic gurus, and finally economics [45].

As an example, the hallmark of neuromuscular dentistry is the use of electronic diagnostic devices in the diagnosis and management of not only orofacial pain patients, but also for discovering problems in asymptomatic patients. Gonzales et al. stated that the results obtained from diagnostic testing should have a high probability of affecting either the correctness of the diagnosis, selection of the appropriate treatment, or both. They found that these electronic diagnostic devices did not provide any of this information [46].

In the American Association For Dental Research’s 3 March 2010 Policy Statement on TMJ Disorders, they state that “…the consensus of recent scientific literature about currently available technological diagnostic devices for TMDs is that except for various imaging modalities, none of them shows the sensitivity and specificity required to separate normal subjects from TMD patients or to distinguish among TMD subgroups.”

Further, this statement goes on to say that neuromuscular dentistry is not specifically recognized by the American Dental Association, although a variety of healthcare providers advertise themselves as TMJ specialists. As a result, many TMD treatments available today are based largely on beliefs, not on scientific evidence.

### 2.10. Problem #10

Misdiagnosis leading to improper and/or inappropriate management has led to iatrogenic disease.

The experience of the past 150 years in the diagnosis and management of chronic orofacial pain conditions has shown that a mechanistic, narrow approach is likely to produce iatrogenic harm, e.g., unnecessary root canal therapy, extractions, bite-changing restorations, TMJ surgery, etc. [47].

In an 18 November 2013 Wall Street Journal article entitled “The Biggest Mistakes Doctors Make”, misdiagnosis was deemed the primary reason for many negative outcomes. As an example, if a clinician believes and espouses the idea that the underlying cause of most TMJ disorders is “internal derangement”, every patient with signs and symptoms will be treated for this so-called “progressive disease”. Since at least 23–35% who go on to an MRI will demonstrate a disc displacement that has nothing to do with their orofacial pain complaint, those findings may persuade the clinician to utilize unnecessary non-invasive, minimally invasive, or, worse yet, invasive treatment, all for the wrong reason. The literature is replete with such cases. Maslow’s comment seems appropriate here: “When you only have a hammer, everything looks like a nail” [48].

## 3. Conclusions

Therefore, based on the above, the following recommendations are offered:The TMJ is a complex orthopedic joint and therefore must be considered part of the body’s musculoskeletal system.The diagnosis and management of TMJ disorders must be based on the same medical orthopedic principles that are used for treating other joints in the body.Clinicians and patients must understand that TMJ disorders are managed, not cured.The etiologic role of synovitis in painful TMJ disorders must be acknowledged and elucidated.Disc displacement is not a disease; instead, it is a non-specific sign of tissue failure leading to biomechanical instability of the joint, which may or may not be clinically significant.TMJ disc function is more important than its position relative to the condyle.Dental occlusion has no direct etiologic relevance to TMJ disorders.Invasive management options must be reserved for patients with demonstrable intra-articular TMJ pathology.Misdiagnosis and misinterpretation of clinical or imaging findings ultimately leads to iatrogenic TMJ problems.The important thing is to never stop questioning. (Einstein, A.)

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
