# Peer review of "Temporomandibular Joint Facts and Foibles"

_jcm, 2023, doi:10.3390/jcm12093246_

Round 1
Reviewer 1 Report
I would like to congratulate the author for the article, it is a very good review, well written and documented. Personally, I liked the conclusions, especially for item #5.
Having said all that, I don’t have any recommendations to give. (Except for removing “?” after the reference number on line 66).
Author Response
Thank you. The ? on line 66 was removed.
Reviewer 2 Report
I would like to congratulate the author on the article which points out the facts and foibles of the diagnosis and management of TMDs. Very interesting article with scientific evidence to support the authors views.
The article points out some common misconceptions of the anatomy of the TMJ, management expectations of TMDs, the role of internal derangement on the development of clinical symptoms, and diagnostic and treatment modalities of TMDs. It is a good review of the current scientific evidence and has debunked some common myths.
The only thing I wanted to point out is the typo in the keywords "Temporomandibu8lar". Otherwise I have no amendments to recommend.
Author Response
Thank you. The correction was made
Reviewer 3 Report
Authors modified the text according to the suggestions.
I found this work impactful and it fits well with in the scope of this journal.
In my opinion, it is suitable for publication.
Author Response
Thank you